# Testing New Coatings for Outdoor Bronze Monuments: A Methodological Overview

Paola Letardi 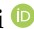

Institute of Anthropic Impacts and Sustainability in the Marine Environment (IAS), CNR, via De Marini 6, 16149 Genova, Italy; paola.letardi@cnr.it

**Abstract:** Coatings to be used for cultural heritage protection face peculiar challenges. In the last few decades, several projects addressed the issue of new treatments in the field of copper alloy artworks. Nonetheless, no one has yet been recognised as a more acceptable solution with respect to traditional choices, with their known limits. The lack of standard methods to test new coatings that can be effectively applied to artworks make it more difficult to compare different studies and open the way to practical use in restoration. Over the years, several issues have gradually been better focused, even though they are not yet widely considered in new coatings efficacy evaluation for application on copper alloy artifacts. They are mainly linked to the quite complex surface of this category of heritage objects and the role it plays on coating effectiveness. An overview of the variety of relevant surface properties is provided (presence of corrosion products and old protective treatments, cleaning methods, surface unevenness, just to name a few) with a special focus on the role of coating performance. Some methodological choices are discussed for the selection of mock-ups, testing techniques and weathering procedures, with peculiar attention to comparison with real artworks.

**Keywords:** metal conservation; coatings efficacy evaluation; atmospheric corrosion; heritage science

## 1. Introduction

Metal objects represents a very broad category in heritage conservation. It is something easy to realise just by having a look at the corrosion identification booklet published by Parks Canada [1] to "provide descriptions and helpful tips, accompanied by photographs, to anyone in charge of metal collections". Selwyn [2] provided a deeper discussion about the known chemical and physical characteristics of "metals and alloys of interest for conservation professionals, along with the different form of corrosion problems indoors, outdoors, and in archaeological settings". In her book, one can find an extensive discussion of metals and alloys of interest for cultural heritage (Table 1), main information on construction steps (Table 2)—which may influence the conservation—and on corrosion basic principles. Specific bibliographic references are provided there for each item.

As a rule, the inherent instability of metallic heritage offers similar preservation challenges to those faced in civil engineering, automotive and construction industries [3]. The main source of this instability is the energy required to extract metals from their ores (smelting), which leaves metals in a high energy state with the tendency to return to the lower energy mineral state [2]. The actual behaviour of every metallic item is the result of complex interactions between the chemical–physical properties of the object and the particular environment around it [4]. These interactions may reach a static or dynamical equilibrium over time. It is thus essential to consider the specific environments all along the lifetime of a metallic heritage object in order to develop treatments and identify realistic conservation goals [4]. Through a deeper understanding of the complex chemical, thermodynamic and kinematic factors, several choices to mitigate the adverse effects of corrosion can be developed by modifying the environment or the surface finishing of the object, as addressed by Corrosion Science [5]. Corrosion is one of the major issues

for metal heritage objects conservation, and methodologies and principles of corrosion science have been slowly entering conservation practice from the second half of the 20th century [3,6–10].

**Table 1.** Most used metals in heritage objects, with common minerals from which they are extracted, historical data and typical appearance (adapted from Tables 1.1, 1.2 and 1.5 in Ref. [2]).

| Metal | Mineral | Formula | Approximate Date of First Widespread Use | Typical Colours of Corrosion Products |
|---|---|---|---|---|
| aluminium | gibbsite | $Al(OH)_3$ | 1800–1900 A.D. (Europe/USA) | colourless or white |
| copper | chalcocite | $Cu_2S$ | ~7000 B.C. (Near East) for native copper ~5000 B.C. (Near East) for smelted copper | Cu(I): red, black, colourless Cu(II): green, blue |
| gold | (native) | Au | 5000–4000 B.C. (Balkans) | – |
| iron | hematite | $Fe_2O_3$ | 1000–0 B.C. (Near East) | Fe(I,III): black Fe(III): red, yellow, orange |
| lead | galena | PbS | 6000–5000 B.C. (Near East/Balkans) | white, red yellow |
| nickel | pentlandite | $(Ni,Fe)_9S_8$ | 2000–1000 B.C. (Near East) for copper/ nickel alloys | green |
| silver | argentite | $Ag_2S$ | 4000–3000 B.C. (Balkans/Near East) | black, white |
| tin | cassiterite | $SnO_2$ | 4000–3000 B.C. (Near East) | black, white |
| zinc | smithsonite | $ZnCO_3$ | 100–200 A.D. (Rome) for copper/zinc alloys 900–1000 A.D. (India) for zinc metal | colourless or white |

**Table 2.** Basic stages of metal objects construction techniques (adapted from Table 1.4 in Ref. [2]).

| Construction Step | Description |
|---|---|
| Forming and Shaping | production by pouring liquid metals into moulds (casting) and by mechanical deformation (forging, rolling; working such as milling, turning, spinning, grinding, stamping, cutting, drilling |
| Assembling | fitting components by welding, soldering, brazing, rivetting, bolting, crinping, gluing |
| Finishing | Completing surface appearance by plating, burnishing, polishing, etching, sand-blasting, painting, lacquering, engraving, chasing, embossing, enameling, patinating |

Although the basic laws underlying corrosion processes in the field of cultural heritage are the same as in industry, the criteria and priorities behind operational choices to mitigate the adverse effects of corrosion are deeply different [4,5,7,11–13]. The vast possibilities of interaction between distinct metals and types of environment give rise to unlimited combinations of corrosion forms, making it useless for conservation purposes to gather occurrences following traditional corrosion classifications. Since the exchange with its surroundings is so intensive, metallic heritage can be better understood and conserved when its context is considered. Accordingly, broad categories of heritage metal objects can be identified depending on the environment where they have been (or still are) [14]:

- Archaeological metals are characterised mainly by long burial in soil [15], water-logged [12,15,16] or underwater [17]; they bear information on very long-term corrosion of metals [18,19]; the equilibrium state reached during burial may be broken when excavated, giving rise to new corrosion process if not properly treated [20].
- A large variety of historic objects (such as scientific instruments, fine arts, historic pieces, ethnographic specimens, etc.) is conserved indoors (museums, monumental buildings, collections); the main preservation strategy in this environment is preventive conservation [4,21], such as humidity control; critical parameter to consider are

dangers from "off-gassing" materials used to build display cases and rooms, as well as air pollution introduced by visitors [21].

- Outdoors monumental and architectural items (sculptures, roofs and decorative objects, functional artifacts and industrial heritage) are mainly subject to weather conditions, pollution and climate change [4,6,14,22–27].

In order to apply correctly the principles of Corrosion Science to the conservation of metal heritage objects, each specific metal or alloy, construction technique, and particular corrosion problem encountered in a museum, outdoor, or archaeological setting should be carefully considered [2–4,15,18,22]. The specific characteristics of heritage objects are the manifold result of a complex and often not completely known history. This can lead to inappropriate treatments when coatings checked for a specific context are adopted in another one without taking into account all the relevant features [28]. For this reason, the topic of protective treatments for metallic cultural heritage can be wide and complex and oversimplified research data may be useless for the conservation of metal heritage objects purposes. Accordingly, from now on the discussion will be restricted to outdoors artifacts. Within this category, the distinctive features relative to bronze sculptures will be considered in order to enlighten the specific characteristics required for conservation treatments [29].

Environmental changes produced by the industrial era raised problems in many fields. Increased pollution and acid rain effects on materials were widely recognised [30] and many research efforts addressed this topic, which is the field of Atmospheric Corrosion [31]. Outdoor copper and copper alloy statues started suffering from a sharp decrease in surface stability and localised corrosion endangering their artistic and aesthetic content [22]. This triggered a growing interaction between conservator-restorers and the scientific community [6–10,25,32]. Over the last decades, this allowed a number of improvements: A growing understanding of the electrochemical processes typical of outdoor bronze monuments [33–39]; the dissemination of the "Theory of Restoration" by Brandi as a tool to lead choices on new conservation practices [7,11,40,41]; more awareness about possible strategies to deal with the changing equilibria and available resources [3,42].

The present understanding of copper and copper alloys atmospheric corrosion ([31], chapters 8, 13, 14; appendix E, K), can be roughly described as the formation of a passivating cuprite layer in clean humid air which evolves into a more complex surface layer (patina) according to the main pollutants present in the surrounding atmosphere [3,25,34]. Green basic copper hydroxysulfates (mainly brochantite) form in $SO_2$ rich atmosphere, while basic hydroxychlorides (atacamite, clinoatacamite) dominate marine environments [43]. Alloying elements play a key role in the corrosion mechanism, which differs from pure copper [34,39,44,45]. A decuprification process was identified [37,44,45], along with cyclic corrosion similar to the "bronze disease" traditionally associated with archaeological copper alloys [2,35,45,46]. The growth of this surface layer on outdoor bronze sculptures and architectonic elements exposed to different weather and pollution conditions is the result of a specific timeline. Several factors determine its local composition and texture, such as the solubility of corrosion products, sheltered and unsheltered exposure, the changing weather (relative humidity, temperature, light and Ultra Violet (UV) radiation, time of wetness), the composition and concentration of air pollutants (Figure 1).

On a single monument, different conditions are often present, and the evolution of the overall result can be detrimental for aesthetic reasons if not for the material loss [42,45]. This leads to the necessity of restoration. The first step consists of cleaning [47–49]. The cleaning methodologies adopted should selectively remove only water-soluble compounds [47], atmospheric particulate deposits, hydrocarbon compounds coming from environmental pollution and other organic compounds from past treatments [50] or failed coatings [51]. At the same time, it should preserve the part of the patina valuable for aesthetic, historical and conservation reasons [48,52]. Afterward, the common practice consists in the use of treatments to prevent or reduce detrimental corrosion, which for outdoor bronze consist in the application of coatings that avoid the contact of the metal/patina layer with the actively corroding agents in the atmosphere (water, corroding ions) and/or other treatments

(inhibitors) to reduce the electrochemical reaction rates [4,5]. Sooner or later, cleaning and application of a protective treatment should be repeated, according to environmental conditions, maintenance programs, etc. (Figure 2).

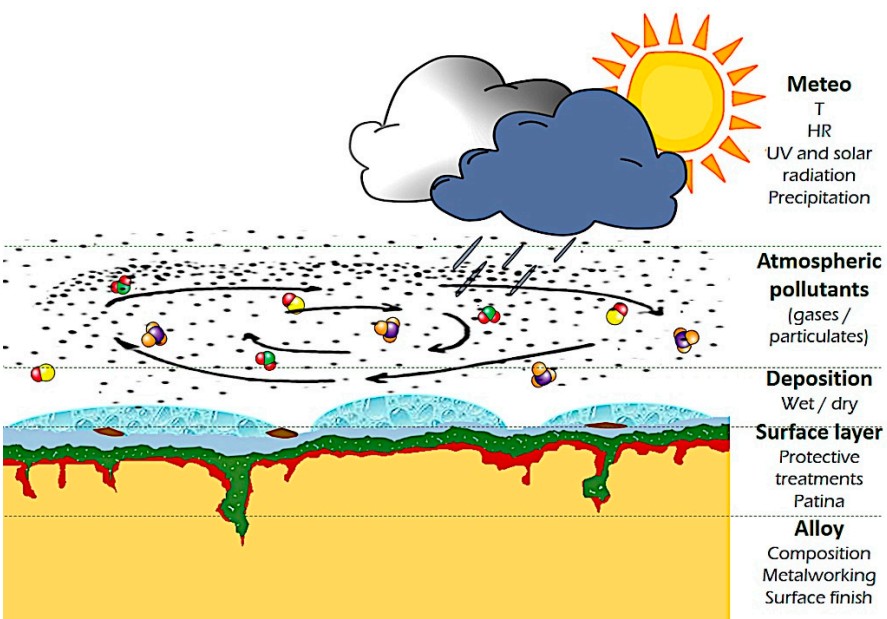

**Figure 1.** Schematic representation of the complex system alloy/surface corrosion layer/protective treatment/environment.

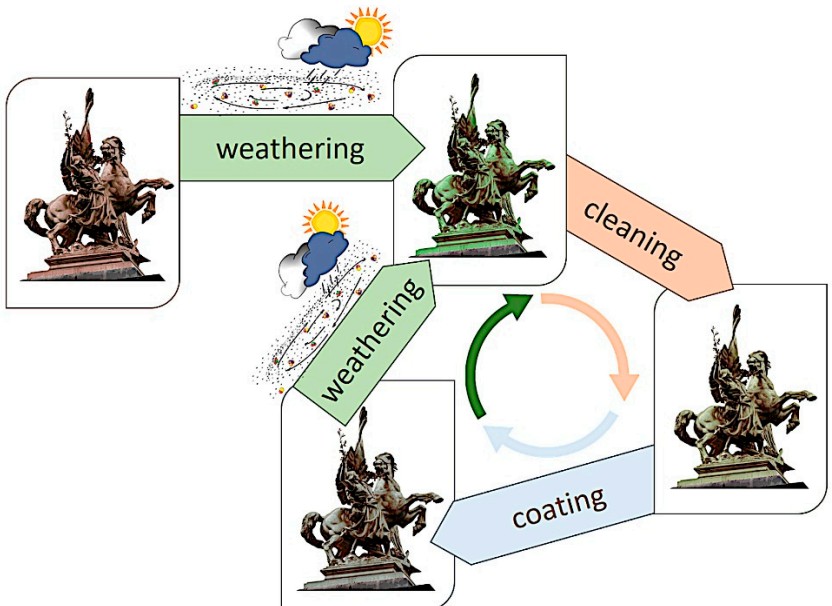

**Figure 2.** Schematic representation of the life cycle of an outdoor bronze monument.

Since the end of the 20th Century, several projects have addressed the need for new/more effective protective treatments to be applied on metallic heritage objects.

The EU-FP3-funded project "New Conservation Methods for Outdoor Bronze Sculptures" [53] considered a new class of sol–gel derived coatings-organic inorganic copolymers, called ormocers (ORganically MOdified CERamics); addressed requirements were the use in outdoor bronze conservation with good protection against corrosion, and at the same time the right compromise between stability and reversibility of the cured coating [54].

After the public engagement following the refurbishment of the Statue of Liberty [55], some sponsors supported the research "Coating strategies for the protection of outdoor bronze art and ornamentation" [56], where the performance of 29 coatings on different copper alloy substrates was addressed and the importance of considering a coating system as a whole, and not only by its part, was pointed out.

In the mainframe of EU-FP6-funded project "EU-ARTECH", traditional treatments performances (Incralac, waxes, Benzotriazole) were compared to innovative treatments ones on coupons with natural and artificial patina [56]; different commercial organo-silanes were tested, the use of limewater was considered to adjust pH toward alkalinity to inhibit bronze disease, and the possible use of fungi to transform unstable corrosion patinas to insoluble copper oxalate was investigated [40]; on copper lamina with natural green patina (mainly brochantite) the Dynasylan F8263 and SIVOClear showed a protective behaviour comparable to Incralac but without perceivable chromatic alterations [57,58]. Biopatina was further investigated in other projects [40,59].

Other EU-funded projects addressed the development of new and more effective protective treatments for outdoor bronze artworks [60,61]. Several efforts were performed to deepen the understanding of the forms and properties of outdoor bronze surface layers [36,39,44,48,62]. Increasing attention was paid to the characterisation and comparison of surface layers and treatments on artworks [63–67]. The literature in this field is very extensive but out of the scope of the present work which is intended to focus on the methodology to be adopted to effectively improve the practice of treatments that conservator-restores can and will apply.

Until now, the transfer of innovative treatments in the practice of restoration of outdoor bronzes was quite poor, despite the considerable amount of laboratory studies in this field since the 1990s. Several methodological problems will be addressed in the following sections to highlight some of the key issues that should be carefully considered to effectively test treatments to improve conservation–restoration practice on a sound scientific base.

## 2. Treatments in Use

It is commonly recognised that the most widely adopted protective treatments on outdoor bronzes are waxes, Incralac and benzotriazole (BTA) [40,68,69].

Incralac was developed in the 1960s as a transparent coating system for polished outdoor copper alloys and minor formulation changes were introduced over the years mostly to comply with environmental regulations [70]. "The coating was trademarked under the name Incralac, but it was never patented and manufacturers have been free to modify the formulation at will" [71]. The performance dependence on chemical patinas, solvent carriers, and additives was also highlighted [56,71]. This caused some difficulties in comparing the several studies devoted to Incralac performance in conservation over the years, as the exact formulation adopted was not always clear and different methodologies, substrates and applications were adopted [70].

A different blend of natural, microcrystalline and polyethylene waxes have been applied on outdoor bronzes [72], according to local use and availability, such as Renaissance wax and Butcher's Boston Polish Amber Paste Wax (microcrystalline and carnauba blended wax) [73], Soter, R21, TeCe Wachs 3534F [63], Paraffin Ozokerite 1899, Microcrystalline wax 1847, Synthetic wax MP-22 [74]. Microcrystalline waxes are by far the most widely accepted solution among conservation professionals. Questions were raised about short lifetime, and reversibility problems [54,56,75] which may strongly depend on the underlying surface and application method [73,74]. Application at room temperature or after heating the surface [74,76], concentration and solvent [76] affect treatment properties such us final thickness, penetration in the patina layer, mechanical properties and corrosion resistance on the same substrate.

BTA is the more widely applied inhibitor for copper alloy heritage objects. It has been used since its appearance on the market, for several decades now [59]. Despite the

huge literature on this topic, some questions still do not have a clear answer [5,70,77]. Its effectiveness on corroded outdoor bronze has been questioned [5,59,78] and issues have been raised on toxicity [5,59].

The "double layer" system, with Incralac and a wax topcoat, was suggested by Marabelli [8], and is since widely adopted [56,62,66].

Several studies have addressed the performance of waxes, Paraloid, Incralac and BTA [28,56–59,63,64,66,67,70–74,76,77,79–82]. The comparison of results among different studies is not straightforward, because each one made different choices on testing methods and analytical and characterisation techniques, coupon alloy and surface properties and application methods; different details on coupon surface properties and application methods are often reported; just a few papers also consider field and weathering studies to some extent [58,63,64,66,67,76,81,82], which are needed to assess coating lifetime. Nonetheless, they highlight issues that deserve greater awareness and attention in the study of these treatments in conservation: they perform differently on clean surfaces and patinated/corroded ones [63,77]; application methods should be considered with greater attention [71,74,76]; the behaviour upon weathering (appearance, barrier properties, degradation) should be better characterised in the specific service conditions [77], along with the easy removal over time [40].

The questionable behaviour of the above-mentioned treatments increased the search for new sustainable and harmless inhibitors and coatings with a better service life to be used on outdoor copper alloys artworks.

## 3. The Ideal Treatment from a Conservation Perspective

From a modern viewpoint, it is nowadays necessary to use eco-friendly and healthy treatments [59,61,83,84].

Treatments to be applied on heritage objects should comply with specific requirements, which go beyond the ones usually considered in other fields of application of metallic objects [56,61].

Among the common recognised goals, treatment should, first of all, stabilise the object (i.e., stop—or significantly slow down—its modification with time) without modifying its visual appearance [5,40,61,85], so only transparent coatings are considered. As some of past treatments applied on artworks show up to be detrimental for their conservation [86], or caused the disfiguration or discolouration of the outer corrosion layers [85], growing attention was paid to the application of new treatments not tested enough and to the principle of minimum invasiveness.

Another requirement concerns the application method: it should be (easily) feasible on the often quite complex surface of outdoor bronze monuments under typical restoration workshop conditions [61,82]. The application/curing temperature has also been suggested to be important for the treatment of large metal sculptures [54,87] where ambient temperature should be the easiest if not the only choice. Application friendly formulations may enhance the quality of the applied film [48]. As inadequate applications can be one of the major sources of failure [70], a detailed easy-to-use application protocol in a restoration workshop may enhance the final effectiveness of the treatment in service.

One of the most discussed requirements is reversibility, i.e., the possibility to be removed easily enough—with safe and healthy methods—leaving the surface it was protecting undamaged [5,54,82]. It is quite important on weathered coatings, as no treatment is supposed to last forever in the continuously changing outdoor conditions. Protective performance (i.e., being tenacious and resistant to adverse environmental conditions), long-term durability (i.e., the possibility for a heritage object to be preserved longer and longer), healthy-cheap and easy maintenance (reversibility and re-applicability) are all widely desired qualities [5,61,82].

Unfortunately, to date, no such treatment exists. Practical restoration activity takes advantage of the growing knowledge both from experience and scientific research. At the same time research activity is moving toward a deeper understanding of conservation needs

in order to properly tailor better treatment solutions. A well-defined balance among the different requirements toward improved solutions would require more detailed knowledge of decay and corrosion routes and rates [3].

## 4. Critical Issues on Testing New Coatings

Standard methods applied in Material Science labs to develop innovative treatments were defined for quite different applications with respect to heritage conservation. To increase the effectiveness of analytical investigations aimed to improve specific outdoor bronze artworks conservation some issues should be considered.

### 4.1. Lack of Reference Standards

Faltermeier addressed the need for a more standardised approach in the testing of inhibitors to treat bronze disease in archaeological copper alloy artifacts. He pointed out how that would improve the reproducibility of results allowing comparison between experimental works [85].

CEN (European Committee for Standardisation) is responsible for planning, development and adoption of European standards. In 2002 the Technical Commission "346 Conservation of Cultural Heritage" (CEN/TC346) was approved with the aim of working on standardisation "in the field of definitions and terminology, methods of testing and analysis, to support the characterisation of materials and deterioration processes of movable and immovable heritage, and the products and technologies used for the planning and execution of their conservation, restoration, repair and maintenance". In 2012, a new Business Plan was set up establishing the definition of standards on a need-based approach [88]. Since 2009, the CEN/TC 346-Conservation of Cultural Heritage published 38 standards. None addresses the specific requirements of testing treatments on metal heritage. Other available standards in coatings evaluation do not reflect the real problems that have to be addressed in this field [89,90]. There are no standardised and shared guidelines that support the assessment of the protectiveness and corrosion behaviour of metallic artworks surface layers [48]. This limits how effectively the design, application and success of conservation treatments can be assessed [3]. The large variety of alloy compositions, surface finishing (i.e., roughness), and patination make it difficult to compare results from different laboratories/projects.

Comparative studies with a selected methodology in a single laboratory allow us to highlight some interesting features: As an example, they made it possible to prove the key role of the different patinas on the performance of the same protective treatment [63,80] and highlighted the critical role of application method on waxes performance [73,74]. A common effort would be advisable to better define effective methodologies to deal with the complexity (Table 3) represented by outdoor bronze conservation, and compare results from different studies.

**Table 3.** Schematic description of the complex system bronze/coating/environment and key features for coatings performance characterisation.

| | Level | Properties | Characteristic Features |
|---|---|---|---|
| Substrate | Alloy | elemental composition<br>metallurgy | principal alloying element<br>casting, rolling, structure |
| | Initial surface finish | polishing, blasting<br>foundry patina | surface roughness<br>colour, composition |
| | Surface layer (patina) | origin<br>composition<br>stratigraphy<br>texture<br>appearance<br>electrochemical properties | natural weathering/accelerated weathering/artificial<br>principal inorganic and organic compounds<br>layers, thickness<br>porosity, morphology, surface roughness<br>colour, gloss<br>corrosion rate |

**Table 3.** *Cont.*

| Level | Properties | Characteristic Features |
|---|---|---|
| Treatment | composition<br>application method<br>texture<br>appearance<br>electrochemical properties | layers, intrinsic features<br>drying time, adhesion, thickness<br>porosity, morphology, surface roughness<br>colour, gloss<br>corrosion rate, inhibition efficacy |
| | weathering stability | UV and light stability<br>stability against condensed moisture<br>water repellent effect<br>coating quality<br>removability/retreatability |
| Exposure condition | rural/marine–rural/<br>industrial/marine–industrial | atmospheric pollutants<br>meteorology<br>precipitation quality<br>wet and dry deposition |

### 4.2. Testing Surfaces

As already described, the surface layer developed as a result of intentional treatments and/or interaction with the environment on outdoor bronze statues should be considered carefully from an aesthetic, historical and conservation point of view. This makes a key difference with industrial application on clean metal/alloy: in most heritage applications, the protective treatments should be applied over pre-existing corrosion products [5,56,91]. Several papers have emphasised the role of surface finishing on the effectiveness of waxes and acrylic coatings used in restoration [63,77,79]. Therefore, new conservation treatments should be tested with a careful definition of environmental context and surface properties [58].

Coupons are needed for extensive analytical studies on new protective treatments, due to the wide homogeneous surface required to compare many different parameters [54,62]. However, making coupons that "look like" the surface structure of outdoor bronze statues—which is the result of past technologies and of complex interactions with the changing environments for many years—may be quite difficult and time-consuming. Often the typical 2–4 years' time scale of project founding forces to make simplified choices.

Simple polished copper coupons were gradually abandoned, with the growing awareness of the role of alloying elements in the bronze corrosion with respect to pure copper [36,62,92]. Several alloys have been used, such as: 88% Cu, 6% Sn, 6% Zn [77]; 88.3% Cu, 5.7% Sn, 1.6% Pb, 3.9% Zn + traces [93]; 91.9% Cu, 2.4% Sn, 1.0% Pb, 2.9% Zn, 0.8% Sb [92]; 88.8% Cu, 2.4% Zn, 4.4% Sn, 3.9% Pb [94]. Sand casting of coupons has been reported [67,77].

Nonetheless, old copper roof tiles from historical buildings are a quick option to test on the natural brochantite patina typical of continental European copper roofs. Thus several laboratories [40,54,56–58,95] have tested treatments on coupons from available tiles (Figure 3a).

Another widely adopted solution to obtain patinated coupons consists of the artificial patination practiced in an artistic foundry. $NH_4Cl$ (chloride patina) [93,96] and the more widely adopted $K_2S$ black patina (Figure 3b) [67,73,77,93,97] were used in different laboratories, with a large spread of surface finishing prior to the artificial patination. The possible role of surface roughness has been suggested [67].

An accelerated ageing technique to obtain pre-corroded surfaces by alternating 12 h artificial rain dropping to 12 h dry until a total Time of Wetness (ToW) of 37 days has been designed in detail. The surface layer of bronze coupon pre-corroded by this "dropping method" is representative of the runoff condition on unsheltered outdoor bronzes with corrosion processes involving decuprification [37,98].

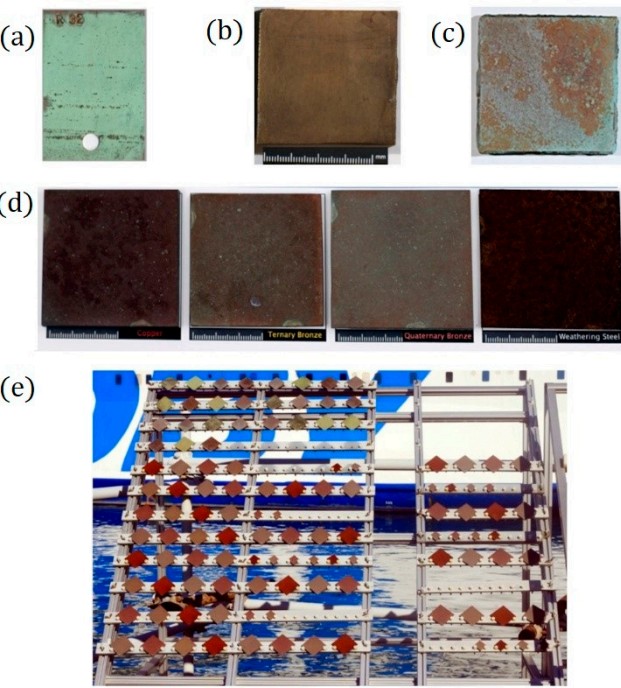

**Figure 3.** Some of the patinated coupon used for treatment testing: (**a**) old copper roof [57]; (**b**) $K_2S$ foundry patina on sand blasted bronze [67]; (**c**) Quaternary Bronze naturally weathered 18 months (initially sprayed; see text) [90] at Genoa Experimental Marine Station (GEMS); (**d**) Copper, Ternary Bronze, Quaternary Bronze and Weathering Steel coupon after 18 m natural weathering [62] at GEMS; (**e**) Exposure rack with copper alloys coupons weathering at GEMS.

Several sets of bronze coupons were naturally aged at a marine Exposure site in Genoa (Figure 3d,e) [99]. A corrosion layer mainly composed of cuprite, atacamite, paratacamite with traces of nantokite and brochantite in some instances was obtained after 12–18 months [62,100]. The detailed composition, thickness, etc. of each set depends on the weather condition during the specific exposure. To enhance the formation of chloride compounds, a group of coupons (Figure 3c) was sprayed twice a week with a 5% NaCl solution for the first 124 days of exposure [90].

The sample size is also an issue to consider [63,85]. While smaller samples may be appropriate when addressing patina growth mechanisms [101], a minimum surface area is required to properly address protective treatment behaviour. Standard panel size adopted in the coating industry was considered too large and 20 mm × 50 mm was suggested as appropriate to enable the treatment of greater numbers of coupons in one experiment, helping the assessment of the reproducibility of experimental results [85]. For the set of coupons exposed to natural weathering in Genoa 60 mm × 60 mm was preferred, with smaller coupons (30 mm × 30 mm) used to characterise the patina growth upon weathering.

### 4.3. Application Methods

The different application methods might produce differences in the treatment properties [5,59,69,74]. Application by brush is widely adopted in conservation workshop [93], along with spraying application [97]. These two methods are then suggested in order to fulfil the requirement of ease of coating application in real practice [61]. Other methods adopted in the coating industry to better control film properties on flat samples may be difficult to compare to outdoor bronze conservation practice.

### 4.4. Ageing Techniques

Natural exposures offer an overall test on treatment effectiveness to cope with the different parameters such as temperature, wet/dry cycles, UV and visible light that influence

copper alloys corrosion [35,38,94]. The time required may be one of the reasons for the scarce literature information that can be found about the natural weathering of coatings on bronze [94]. Nonetheless, they can be fundamental to properly characterise how long they last and properly rank the comparison of different treatments [63,67].

Artificial tests are quicker and necessarily consider only a limited number of variables. Climatic chamber exposure with temperature/UV cycles [97] and the dropping method [61] were reported.

### 4.5. Analytical Techniques

Getting data on the specific corrosion of heritage metals is a great challenge [3]. This is especially true for outdoor bronze monuments, which have large and uneven surfaces. The availability of portable, in situ Non-destructive Techniques greatly help a wider characterisation of artworks and allow a straightforward comparison of results on coupons and on heritage metals [67,90]. As corroded bronze surfaces may have quite an uneven appearance, the use of frames (Figure 4) to re-position the measurement area of the instruments in a precise manner is strongly advised for monitoring purposes.

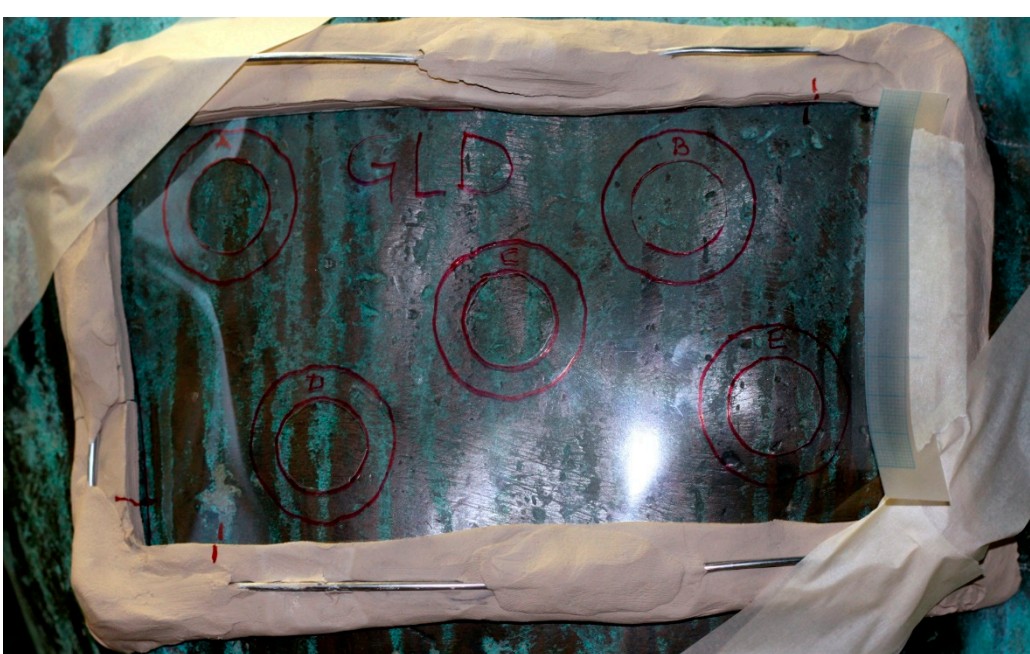

**Figure 4.** Frame used to reposition measurement instruments on a selected surface for monitoring purpouses. A number of different areas are identified and labeled (A to D) for statistical reasons.

Over the years more and more analytical techniques have been used to characterise heritage objects surfaces [102], with the increasing availability of portable Non-destructive Techniques (NdT) [103–105].

In order to characterise the different properties of patinas and conservation treatments on outdoor bronze monuments, a multi-analytical approach is required [51–67]. Visual appearance, chemical composition, corrosion rate and surface texture are among the relevant parameters to consider. Colour measurements are quite useful to monitor the aesthetic appearance and quantify colour differences [48,51,61]. Portable digital microscopes have been used for in situ documentation of surface morphology [48,67]. Electrochemical techniques are quite effective to characterise barrier properties of organic coatings [104] and their use was growingly adopted in the last decades [66,72,106]. The development of a special setup for NdT testing in situ on outdoor bronzes has also highlighted the role in addressing patina properties. Fourier-transform infrared spectroscopy (FTIR) in reflectance mode has proved to be useful to analyse both coatings and patina chemical composition both on coupons and in field measurements [51,67,93]. Portable near-mid-infrared (NIR-MIR) total

reflection analysis was recently investigated [105]. Among others, portable surface roughness measurements [67], Raman and X-Ray Fluorescence Spectroscopy (XRF) [103] are also valuable techniques to collect the different information required to better understand the effectiveness of protective treatments on the complex outdoor bronze surfaces and plan better conservation practices.

## 5. Final Comments

The need for "a dialogue among conservators, curators, environmental scientists, and corrosion engineers" to solve the puzzle of preserving bronze sculpture in an outdoor environment was recognized long ago [7]. Over the years, a deeper understanding of the quite complex corrosion of outdoor bronze surfaces in the changing environment has emerged, along with a better understanding of the values of the different kinds of surface products which also moved the perspectives of restoration–conservation choices [22,29,91]. A wider awareness of present knowledge and on the challenges that have to be addressed may help to employ the few resources available more effectively.

Analytical developments and portable NdTs [102,103] can greatly improve our understanding of the corrosion mechanism of outdoor bronze artworks and outline a deeper case by case comprehension to better define conservation procedures. Notably, a comparison of measurements on coupons and on heritage surfaces can help highlight the principal factors to be considered when preparing mock-ups for treatment development and testing. The choice of representative coupons for treatment testing deserves some more effort toward the best balance between time and representative surface structure. Other challenges remain to be tackled. Cooperation among the different people involved in these activities [3,11] may help to improve experimental design and a common effort would be desirable in drafting guidelines toward more comparable results. Greater attention to ageing protocols would be also advisable in this field. Platforms for sharing data [103] would help to improve follow-up to past projects. All this may help to really develop more effective conservation methodologies for society.

**Funding:** This research received no external funding.

**Institutional Review Board Statement:** Not applicable.

**Informed Consent Statement:** Not applicable.

**Data Availability Statement:** Data sharing not applicable.

**Acknowledgments:** The frame shown in Figure 4 was made for a diagnostic activity during a restoration project in collaboration with B.Salvadori (INPC-CNR) and N.Salvioli (restorer).

**Conflicts of Interest:** The author declares no conflict of interest.

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
