# Peer review of "Testing New Coatings for Outdoor Bronze Monuments: A Methodological Overview"

_coatings, doi:10.3390/coatings11020131_

Round 1

Reviewer 1 Report

The manuscript is a mini-review that deals with a topic of interest in the field of conservation of bronze art objects in the outdoor environment.

Observations and suggestions

1. The paper is quite well written, but there are still mistakes of writing, of expression in English, so the material should be reread by the author to eliminate small mistakes and by a native English speaker.

2. This review should contain the latest publications with achievements on this topic, but the references are not enough even for a minireview.

3. The author refers here to a table (table 1), which belongs to the reference [40] since there is no table in the manuscript. This issue needs to be clarified! (Page 3 Line 114). My opinion is that a table containing a comparison between the different strategies to check protective effectiveness would be beneficial for readers and the author should take this into account to increase the quality of the study.

4. The same observation for the section of “4.5. Analytical techiniques”. Here a comparative table on the advantages and disadvantages of analytical methods in the literature would be beneficial for the reader. However, this section needs to be developed because it currently contains only a list of methods, without comparative details and discussions.

5. There are no figures in this review. This is not so good for supporting a review article. Some impact figures that cover the chosen topic should be included in this review. My opinion is that for a high scientific quality review each Figure should include several small pictures each being from other work in order to cover all the information presented in the literature.

6. Comparison with other similar reviews should be addressed and discussed.

7. Also, the "Final comments" section should be rewritten and should become a critical essay of the author on the topic addressed. This section seems to me to be treated superficially

8. By summing up, this contribution cannot be recommended for publication in this form. Author is advised to carefully revise and complete the information presented.

Author Response

Observations and suggestions

    1. The paper is quite well written, but there are still mistakes of writing, of expression in English, so the material should be reread by the author to eliminate small mistakes and by a native English speaker.

done

    2. This review should contain the latest publications with achievements on this topic, but the references are not enough even for a minireview.

Not many references address the topic of methodology to be adopted; many references have been added to better describe the complexity of the system under examination, and several examples of different methods adopted

    3. The author refers here to a table (table 1), which belongs to the reference [40] since there is no table in the manuscript. This issue needs to be clarified! (Page 3 Line 114). My opinion is that a table containing a comparison between the different strategies to check protective effectiveness would be beneficial for readers and the author should take this into account to increase the quality of the study.

Reference to the table in (former) ref 40 has been removed (it listed different papers with EIS measurements on waxes and Incralac); a more detailed description of the different waxes tested, and results on application methods have been added.

   4. The same observation for the section of “4.5. Analytical techiniques”. Here a comparative table on the advantages and disadvantages of analytical methods in the literature would be beneficial for the reader. However, this section needs to be developed because it currently contains only a list of methods, without comparative details and discussions.

Reference to available review on analytical techniques in Heritage Science have been added. The discussion has been focused on the need of a multi analytical investigation in order to consider as a whole the manifold requirements to be satisfied in this field.

    5. There are no figures in this review. This is not so good for supporting a review article. Some impact figures that cover the chosen topic should be included in this review. My opinion is that for a high scientific quality review each Figure should include several small pictures each being from other work in order to cover all the information presented in the literature.

Four figures (and three tables) have been added to draw the attention on

    6. Comparison with other similar reviews should be addressed and discussed.

To the best of my knowledge, there is no other methodological overview which try to focus on details of the system “outdoor bronze” (cfr tab.3 ) as a whole

    7. Also, the "Final comments" section should be rewritten and should become a critical essay of the author on the topic addressed. This section seems to me to be treated superficially

The section has been revised, with a summary of the critical points focused in the previous sections

    8. By summing up, this contribution cannot be recommended for publication in this form. Author is advised to carefully revise and complete the information presented.

Thank you for your comments and suggestions. I hope the revision has taken them properly into account

Reviewer 2 Report

The paper “Testing new coatings for outdoor bronze monuments: a methodological overview”, by Paola Letardi, present interesting details related to new coatings for monuments exposed to Ambiental pressure.

The paper is recommended for publication after major revisions:

  1. As a first general remark: the current paper represents a Review, thus the Introduction must be reconsidered; is too small. Also, the references are not so many (only 66). Please be aware that ordinary articles have the same Introduction size like your Review. The author is asked to increase to Introduction size, by adding more references from the literature.
  2. A second general remark: please insert a space after each comma when you have many references inserted in the same brackets.
  3. Line 26: please remove the apostrophe from “to” word.
  4. Line 27: please remove the apostrophe from “collections” word.
  5. Line 28: please remove the apostrophe from “of” word.
  6. Line 30: please remove the apostrophe from “settings” word.
  7. Line 40: please use the superscript for “20th” word.
  8. Line 52: please use subscript for “SO2” formula.
  9. Line 83: please use the superscript for “20th” word.
  10. Line 96: please remove the apostrophe from “The” word.
  11. Line 97: please remove the apostrophe from “will” word.
  12. Line 167: please remove the apostrophe from “in” word.
  13. Line 170: please remove the apostrophe from “maintenance” word.
  14. Line 198: please use subscript for “NH4Cl” and “K2S” formula.
  15. Line 214: please use the superscript for “mm2” unit word.
  16. Line 216: please use the superscript for “mm2” unit word.
  17. Line 217: please use the superscript for “mm2” unit word.

Author Response

1. As a first general remark: the current paper represents a Review, thus the Introduction must be reconsidered; is too small. Also, the references are not so many (only 66). Please be aware that ordinary articles have the same Introduction size like your Review. The author is asked to increase to Introduction size, by adding more references from the literature.

The introduction has been extended, and several references added to better describe the scenario in which coatings for outdoor bronze should work and introduce the discussion of critical aspects to be considered. 

 2. A second general remark: please insert a space after each comma when you have many references inserted in the same brackets.

done   

   3. Line 26: please remove the apostrophe from “to” word.

    4. Line 27: please remove the apostrophe from “collections” word.

    5. Line 28: please remove the apostrophe from “of” word.

    6. Line 30: please remove the apostrophe from “settings” word.

I apologise for using ‘ instead of “; the text was taken from the cited reference, and quotes was believed appropriate; I hope that turning apostrophe to quotation mark is ok.

    7. Line 40: please use the superscript for “20th” word.

done

    8. Line 52: please use subscript for “SO2” formula.

done

    9. Line 83: please use the superscript for “20th” word.

done

   10. Line 96: please remove the apostrophe from “The” word.

   11. Line 97: please remove the apostrophe from “will” word.

As above; I hope that turning apostrophe to quotation mark is ok.

    12. Line 167: please remove the apostrophe from “in” word.

   13.  Line 170: please remove the apostrophe from “maintenance” word.

As above; I hope that turning apostrophe to quotation mark is ok.

    14. Line 198: please use subscript for “NH4Cl” and “K2S” formula.

done

   15. Line 214: please use the superscript for “mm2” unit word.

   16. Line 216: please use the superscript for “mm2” unit word.

    17. Line 217: please use the superscript for “mm2” unit word.

done

Round 2

Reviewer 1 Report

I have carefully read the revised version of the manuscript and the author's answer and I consider that the figures, tables, references and comments introduced have greatly increased the quality of the work. In this form, I am pleased to recommend  the manuscript to be published in Coatings journal.

Reviewer 2 Report

The paper “Testing new coatings for outdoor bronze monuments: a methodological overview”, by Paola Letardi, present interesting details related to new coatings for monuments exposed to ambiental pressure. The author made significant improvements in order to rise the scientific quality of the paper. The review is very well written, and the information is clearly presented. 

The paper is recommended for publication in the present form.